# Physicochemical characteristics and microbial community succession during oat silage prepared without or with *Lactiplantibacillus plantarum* or *Lentilactobacillus buchneri*

Yanzi Xiao,[1,2] Lin Sun,[3] Xiaoping Xin,[2] Lijun Xu,[2] Shuai Du[4]

**ABSTRACT**   This study aimed to investigate the effects of *Lactiplantibacillus plantarum* (*L. plantarum*) or *Lentilactobacillus buchneri* (*L. buchneri*) on the fermentation characteristics and microbial community of oat silages in response to additives. The oat was harvested at the milk stage of maturity and was chopped into 30 mm in size. Then, the oat was treated with distilled water (control, CON treatment), *L. buchneri*, and *L. plantarum*; the addition of *L. buchneri* and *L. plantarum* was in $1 \times 10^6$ colony-forming units/g of fresh matter and stored at room temperature (25°C). Results showed that the addition of *L. plantarum* inoculations could increase the lactic acid concentrations of oat silages compared with the control, and the addition of *L. buchneri* could increase the acetic acid concentrations, whereas the addition of *L. plantarum* could decrease the fiber contents and increase the crude protein content. The Shannon index of bacterial community was markedly ($P < 0.05$) lower in the *L. buchneri*- and *L. plantarum*-treated oat silages, and the Shannon index of fungal community was significantly ($P < 0.05$) higher in *L. buchneri*- and *L. plantarum*-treated oat silages compared with the CON treatment for 7 and 10 days of ensiling. From 7 to 90 days of ensiling, *Lactobacillus* was the dominant genus during the whole fermentation process in the three treatments. The homofermentative *L. plantarum* regulated the fermentation quality and microbial community by enhancing the Emden–Meyerhoff pathway, phosphoketolase pathway, and pentosephosphate pathway, and the heterofermentative *L. buchneri* modulated the ensiling performance and microbial community via improving the pentosephosphate pathway. These results suggested that the addition of lactic acid bacteria could improve the ensiling performance by regulating the microbial community in oat silage, and *L. plantarum* was more beneficial than *L. buchneri* for enhancing the fermentation quality.

**IMPORTANCE**   Ensiled whole-plant oats are an important feedstuff for ruminants in large parts of the world. Oat silage is rich in dietary fibers, minerals, vitamins, and phytochemicals beneficial to animal health. The fermentation of oat silage is a complex biochemical process that includes interactions between various microorganisms. The activity of many microbes in silage may cause an extensive breakdown of nutrition and lead to undesirable fermentation. Moreover, it is difficult to make high-quality oat silage because the number of epiphytic lactic acid bacterium microflora was lower than the requirement. Understanding the complex microbial community during the fermentation process and its relationship with community functions is therefore important in the context of developing improved fermentation biotechnology systems. These results suggested that the addition of *Lactobacillus plantarum* or *Lactobacillus buchneri* regulated the ensiling performance and microbial community in oat silage by shaping the metabolic pathways.

**KEYWORDS**   lactic acid bacteria, microbial community, fermentation quality, oat silage

Address correspondence to Lijun Xu, xulijun@caas.cn, or Shuai Du, dushuai_nm@sina.com.

The authors declare no conflict of interest.

Forage oat (*Avena sativa*) is considered one of the most significant feed resources worldwide due to its high yield, rich nutritional content, and adaptability to diverse environments. It has become an essential component of livestock feed and is widely used in the livestock industry (1, 2). Ensiling is a traditional, essential, and dependable technique for preserving forages and crops, which has played a vital role in the development of animal husbandry. In recent years, there has been a growing focus on ensiling, particularly in developed countries (3, 4). The process of ensiling has been demonstrated to generate organic acids, primarily lactic acid (LA), through the microbial breakdown of water-soluble carbohydrates (WSCs). This creates an anaerobic environment with increased acidity, which helps inhibit the growth of spoilage microorganisms (5, 6). The preservation of silage from forages and crops depends on the microbial ecology present. Lactic acid bacteria (LAB) are a crucial component of the microbiome and play a vital role in ensuring the production of high-quality silage. Moreover, LAB have a crucial potential to modulate the microbiome and the end-products during the anaerobic fermentation process, especially in the silage additives currently available (7).

The application of LAB inoculants has been extensively used to enhance the retention of forage nutrition and improve ensiling performance (8). The effectiveness of LAB inoculants is dependent on their growth performance and their ability to induce rapid acidification, increase aerobic stability, and enhance the digestibility of silages (9). LAB can be classified into three categories: obligately homofermentative, obligately heterofermentative, and facultatively heterofermentative according to the tolerance of LAB to oxygen and the end-products of fermented carbohydrates (10). Prior research studies have suggested that heterofermentative LAB, such as *Lentilactobacillus buchneri* (*L. buchneri*), produce a combination of lactic acid, acetic acid (AA), and ethanol through the Phosphoketolase pathway, resulting in improved aerobic stability (8, 10, 11), while homofermentative LAB, such as *Lactiplantibacillus plantarum* (*L. plantarum*), can utilize glucose to obtain LA through the Emden–Meyerhof pathway to accelerate the fermentation process. Despite the potential benefits of LAB inoculants, not all of them are effective in significantly improving silage quality. This is because of the diversity of LAB inoculants, making it challenging to create LAB-based products that work effectively in all situations (11, 12). Although the fermentation process did not have a significant impact on the alteration of bacterial and fungal communities, these microorganisms were still linked to the quality of silage (13). The dynamics of the bacterial community were also investigated worldwide in alfalfa, corn, barley, wheat (14–17), and other silages for agriculture, while the research on the alternations of the fungal community during the fermentation process has been limited.

The objective of this study was to evaluate the effectiveness of bioaugmentation using *L. plantarum* and *L. buchneri* on ensiling performance, including physicochemical and fermentation characteristics, as well as the microbiome of oat silage. Furthermore, the study aimed to investigate the correlations between the physicochemical and fermentation characteristics and the microbiome of the silage.

## MATERIALS AND METHODS

### Substrate and silage

Forage oat (*Avena sativa* L.), obtained from the Hulunber Grassland Ecosystem National Observation and Research Station of the Chinese Academy of Agricultural Sciences in Inner Mongolia, China, was harvested at the milk stage of maturity. Then, the oat was wilted for 3 h after harvest under the sunny condition (25°C–27°C). The samples were collected from three randomly selected sites. The oat was chopped into 30 mm in size using a forage cutter (Fulida Tool Co. Ltd., Linyi, China) and brought to the laboratory immediately. Three randomly selected sites were chosen, and for each site, there were 12 forage piles (one untreated and two inoculated piles for each fermentation time of 7, 10, 60, and 90 days). The oat piles were treated separately using different methods, including distilled water (control, CON), *L. buchneri* (LB, $1 \times 10^6$ colony-forming units

(cfu)/g of fresh matter, purchased from Jiangsu Lvke Biotechnology Company, Gaoyou, China), and *L. plantarum* (LP, $1 \times 10^6$ colony-forming units/g of fresh matter, purchased from Jiangsu Lvke Biotechnology Company, Gaoyou, China). The chopped oat, treated with or without inoculants, was packed into vacuum-sealed polyethylene plastic bags (260 mm × 180 mm) and stored at room temperature (25℃). Each treatment group consisted of three replicates with approximately 250 g of forage per bag. Samples were collected and analyzed after 7, 10, 60, and 90 days of ensiling. The analysis included measuring the chemical composition, fermentation quality, and microbiome of the oat silage.

## Fermentation characteristics and chemical composition analyses

For sampling of chemical composition parameters, clean containers were used to collect fresh materials (FM) and oat silage after being uniformly blended. The dry matter (DM) content was determined by drying a subsample in an oven for 72 h at 65℃ and then grinding it through a 1-mm screen (FW100, Taisite Instrument Co. Ltd., Tianjin, China) for further chemical analysis. The ANKOM A200i Fiber Analyzer (ANKOM Technology, Macedon, NY, USA) was utilized to determine the fiber compositions, including the lignin, neutral detergent fiber (NDF), and acid detergent fiber (ADF) contents, following previous reports (18, 19). The crude protein (CP) content was determined using the method of the Association of Official Analytical Chemists (20). The anthrone method was utilized to determine the water-soluble carbohydrate content (21). Ten grams of silage samples was mixed with 90 mL sterile water and stored at 4℃ fridge for 24 h for the extractions. After extraction, the samples were filtered through four layers of cheesecloth. The pH of the filtrate was measured using a glass-electrode pH meter. The concentrations of organic acids [lactic acid, acetic acid, propionic acid (PA), and butyric acid (BA)] in the oat silage were determined by the high-performance liquid chromatography according to the previously reported method (22). The ammonia nitrogen ($NH_3$-N) content was determined using the phenol–hypochlorite method based on the previously published method (23). The microbial populations, including LAB, yeasts, and molds, in the FM were counted using the plate count method and expressed as cfu/g of FM (22). The de Man, Rogosa, Sharpe agar (Difco Laboratories, Detroit, MI, USA) was used to count the numbers of LAB after incubating at 30℃ for 48 h. The potato dextrose agar (Nissui Ltd., Tokyo, Japan) was used to count the numbers of molds and yeast after incubating at 30℃ for 48 h.

## DNA extraction, PCR, and sequencing

To analyze the microbiome of the oat silage, all samples were initially stored at −80℃. The microbial DNA was extracted from both the FM and silage samples using HiPure Stool DNA Kits (Magen, Guangzhou, China) following the manufacturer's instructions. The bacterial community analysis involved the production of amplicons covering the V5–V7 hypervariable regions of the 16S rRNA gene, using primers 799F (5′-AACMGGATTA-GATACCCKG-3′) and 1193R (5′-ACGTCATCCCCACCTTCC-3′) as described by Beckers et al. (24). Additionally, primers ITS1_F_KYO2 (5′-TAGAGGAAGTAAAAGTCGTAA-3′) and ITS86R (5′-TTCAAAGATTCGATGATTCAC-3′) were used to target the ITS region of fungal DNA for amplicon production, as reported by Scibetta et al. (25). The PCR amplification was performed in a total volume of 25 µL reaction mixture, including template DNA (25 ng), PCR premix (12.5 µL), and primer (2.5 µL of each primer), and the volume was adjusted with PCR-grade water. The PCR conditions to amplify the prokaryotic 16S fragments consisted of an initial denaturation for 30 s (98℃); 35 cycles of denaturation for 10 s (98℃), annealing for 30 s (54℃ /52℃), and extension for 45 s (72℃); and then final extension for 10 minutes (72℃). The PCR products were performed with 1% agarose gel electrophoresis. In the DNA extraction process, ultrapure water was used to exclude the possibility of false-positive PCR results as a negative control. The PCR products were purified using AMPure XT beads (Beckman Coulter Genomics, Danvers, MA, USA) and quantified using Qubit (Invitrogen, USA). The libraries were sequenced either on 300PE

MiSeq runs, and one library was sequenced with both protocols using the standard Illumina sequencing primers, eliminating the need for a third index read.

## Microbial community analyses

The editing of reads, selection of unique sequences, identification of chimeras, assembly of reads, and determination and taxonomic classification of amplicon sequence variants (ASVs) were performed using the Divisive Amplicon Denoising Algorithm (DADA2) in R version (3.5.1) (26). The ASVs were then taxonomically classified using a naïve Bayesian model and the RDP classifier (http://rdp.cme.msu.edu/) based on either the SILVA database (for bacteria) reported by Pruesse et al. (27) or the ITS2 database (for fungi) described by Ankenbrand et al. (28), with a confidence threshold value of 80%. Alpha diversity metrics, including the Chao1 index and Good's coverage, were computed using QIIME (version 1.9.1) (29). The principal coordinates analysis (PCoA) was generated using either unweighted or weighted UniFrac distance in the R package (version 2.5.3). The abundance statistics for each taxonomic group were obtained using Krona (version 2.6) (30). The bacterial community composition was obtained using the R package (version 2.2.1) (31), and the results were presented in a stacked bar plot. The abundance of phyla and genera for the fungal community was graphed using Circos (version 0.69–3) (32), and the results were displayed using a circular layout. The pheatmap package (version 1.0.12) was used to plot the abundance of genus and visualize the results using heatmap (33). The R package (version 1.8.4) was used to calculate Pearson correlation analysis of species, and the network of correlation coefficient was generated using an online platform (http://www.omicsmart.com) (34). An online tool (https://www.omicstudio.cn/tool/60) was utilized to perform the linear discriminant analysis (LDA) effect size (LEfSe) analysis, using a threshold of LDA score > 3 and $P < 0.05$.

## Statistical assessments

The data analyzed for DM, chemical composition (CP, WSC, ADF, NDF, and lignin content), and ensiling performance (pH, LA, AA, PA, and $NH_3$-N parameters) were presented as the mean ± standard error of the means for three replicates. The effects of additives and fermentation period on silage quality were assessed using SAS 9.0.

All data were analyzed using general linear models with the following equation: $Y_{ij} = \mu + \alpha_i + \beta_j + (\alpha \times \beta)_{ij} + \varepsilon_{ij}$, where $Y_{ij}$ represents the response variable, $\mu$ is the overall mean, $\alpha_i$ represents the effect of the additive, $\beta_j$ represents the effect of the ensiling period, $(\alpha \times \beta)_{ij}$ represents the interaction effect between the additive and ensiling period, and $\varepsilon_{ij}$ represents the residual error and the significant differences among treatments were at the 5% probability level (35). The correlations between the variables and the factors (additives and fermentation period) were evaluated using this model (6).

## RESULTS

### Characteristics of the forage prior to ensiling

The nutritional and microbial compositions of oat are shown in Table 1. The contents of WSC, CP, ADF, and NDF of oat were 16.30%, 12.46%, 34.05%, and 53.33% of DM, respectively. The oat contained low desirable LAB (4.57 log cfu/g of FW) and high mold (4.13 log cfu/g of FW) counts.

### Chemical compositions of ensiled oat

The nutritional compositions of oat silage are enumerated in Table 2. The WSC, CP, ADF, NDF, and lignin contents were significantly ($P < 0.05$) influenced by the effects of inoculants or ensiling days. The interaction between inoculants and ensiling days significantly ($P < 0.05$) affected the WSC, CP, ADF, NDF, and lignin. No significant ($P > 0.05$) differences were observed in the DM content in all treatments, while the reduction of the WSC content along with ensiling process was present throughout all treatments.

**TABLE 1** Chemical and microbial compositions of oat prior to ensiling[a]

| Item | Oat |
|---|---|
| Dry matter (%) | 42.69 ± 1.26 |
| Water soluble carbohydrate (% DM) | 16.30 ± 0.36 |
| Crude protein (% DM) | 12.46 ± 0.12 |
| Neutral detergent fiber (% DM) | 53.33 ± 1.57 |
| Acid detergent fiber (% DM) | 34.05 ± 0.71 |
| Lignin (% DM) | 4.81 ± 0.11 |
| Lactic acid bacteria (lg cfu/g FW) | 4.57 ± 0.07 |
| Yeasts (lg cfu/g FW) | 4.67 ± 0.11 |
| Molds (lg cfu/g FW) | 4.13 ± 0.13 |

[a]DM, dry matter; FW, fresh weight.

For 90 days of fermentation, the LB treatment exhibited lower WSC content compared with both the CON and LP treatments. The CP content in all silages decreased during the ensiling process. The LP treatment had a higher CP content compared with the CON or LB treatments. There were no significant ($P > 0.05$) differences in the ADF concentration in the CON and LP treatments, whereas the ADF concentration was significantly ($P < 0.05$) decreased in the LP treatment compared with that in the CON and LB treatments after 90 days of fermentation. Interestingly, the alternations of NDF concentration were in accordance with the ADF content. The NDF content was significantly ($P < 0.05$) decreased in the LP treatment compared to that in the CON and LB treatments, and the ADF content in CON was higher than that of the *L. buchneri*-inoculated silage. Throughout the fermentation process, the LP treatment showed a noteworthy reduction in lignin

**TABLE 2** Chemical compositions of ensiled oat[a]

| Item | Day | Treatment | | | SEM | P-value | | |
|---|---|---|---|---|---|---|---|---|
| | | CON | LB | LP | | Inoculants | Fermentation | Interaction |
| DM (g kg⁻¹ DM) | 7 | 42.83ᵃᴬ | 42.56ᵃᴬ | 42.50ᵃᴬ | 0.34 | 0.148 | 0.082 | 0.265 |
| | 10 | 43.16ᵃᴬ | 42.60ᵃᴬ | 42.33aᴬ | | | | |
| | 60 | 43.14ᵃᴬ | 43.13ᵃᴬ | 42.22ᵃᴬ | | | | |
| | 90 | 41.85ᵃᴬ | 43.05ᵃᴬ | 42.32ᵃᴬ | | | | |
| WSC (g kg⁻¹ DM) | 7 | 2.57ᵃᴬ | 2.83ᵃᴬ | 2.80ᵃᴬ | 0.12 | 0.001 | <0.001 | 0.007 |
| | 10 | 1.86ᵇᴮ | 2.40ᵃᴬᴮ | 3.00ᵃᴬ | | | | |
| | 60 | 1.76ᵃᴮ | 2.43ᵃᴬᴮ | 2.36ᵃᴮ | | | | |
| | 90 | 2.15ᵃᴬᴮ | 2.00ᵃᴮ | 2.16ᵃᴮ | | | | |
| CP (g kg⁻¹ DM) | 7 | 12.57ᵇᴬ | 13.00ᵇᴬ | 13.75ᵃᴬ | 0.44 | <0.001 | 0.958 | 0.003 |
| | 10 | 11.20ᵇᴮ | 12.79ᵃᴬ | 14.03ᵃᴬ | | | | |
| | 60 | 11.46ᵇᴬᴮ | 12.28ᵇᴬ | 14.08ᵃᴬ | | | | |
| | 90 | 12.45ᵇᴬᴮ | 12.25ᵃᵇᴬ | 13.42ᵃᴬ | | | | |
| ADF (g kg⁻¹ DM) | 7 | 41.37ᵃᵇᴬ | 42.50ᵃᴬ | 39.70ᵇᴬ | 1.04 | <0.001 | 0.161 | 0.026 |
| | 10 | 42.94ᵃᴬ | 40.68ᵇᴮ | 39.55ᵇᴬ | | | | |
| | 60 | 43.26ᵃᴬ | 42.75ᵃᴬ | 39.62ᵇᴬ | | | | |
| | 90 | 42.08ᵃᴬ | 42.40ᵃᴬ | 40.34ᵇᴬ | | | | |
| NDF (g kg⁻¹ DM) | 7 | 66.97ᵃᴬᴮ | 68.00ᵃᴬ | 63.23ᵇᴬ | 1.39 | <0.001 | 0.613 | 0.008 |
| | 10 | 68.40ᵃᴬ | 66.68ᵇᴬᴮ | 62.63ᶜᴬ | | | | |
| | 60 | 67.54ᵃᴬᴮ | 66.38ᵃᴮ | 62.62ᵇᴬ | | | | |
| | 90 | 66.08ᵃᴮ | 65.58ᵃᴮ | 63.11ᵇᴬ | | | | |
| Lignin (g kg⁻¹ DM) | 7 | 5.49ᵃᵇᴬ | 5.79ᵃᴬ | 5.06ᵇᴬ | 0.05 | <0.001 | 0.426 | 0.043 |
| | 10 | 5.66ᵃᴬ | 5.39ᵃᵇᴮ | 5.10ᵇᴬ | | | | |
| | 60 | 5.65ᵃᴬ | 5.77ᵃᴬ | 5.14ᵇᴬ | | | | |
| | 90 | 5.42ᵃᴬ | 5.70ᵃᴬᴮ | 5.32ᵇᴬ | | | | |

[a]Lowercase letters (a–c) were used to indicate significant differences among the treatments on the same ensiling days at $P < 0.05$. Capital letters (A–B) were used to indicate significant differences within the same treatment on the same ensiling days at $P < 0.05$. DM, dry matter; WSC, water-soluble carbohydrate; CP, crude protein; ADF, acid detergent fiber; NDF, neutral detergent fiber; CON, control; LB *Lentilactobacillus buchneri*; LP, *Lactiplantibacillus plantarum*; SEM standard error of means.

content, a difference that was statistically significant ($P < 0.05$) when compared with both the CON and LB treatments. Interestingly, the lignin content in the CON was generally lower than that in the LB treatment, except for the 10-day ensiling period.

## Fermentation of ensiled oat

The fermentation characteristics of oat silage were analyzed, and the results are presented in Table 3. The LA, AA, PA, and $NH_3$-N contents and pH value were significantly ($P < 0.05$) influenced by the effects of inoculants or ensiling days. The interaction between inoculants and ensiling days significantly ($P < 0.05$) affected the LA, AA, and $NH_3$-N contents. It became evident that the pH levels decreased uniformly across all treatments as the fermentation progressed. Nevertheless, it's noteworthy that the LB treatment was consistently maintained a slightly higher pH when contrasted with both the CON and LP treatments. As the ensiling process unfolded, the pH levels continued to drop across all treatments, with the LP treatment consistently displaying a lower pH compared with the CON treatment. The LA content increased continuously in all silages throughout the fermentation period. Notably, the LB treatment had a lower LA content compared with the CON and LP treatments after 7, 10, 60, and 90 days of ensiling. In contrast, the AA content showed an increase during ensiling in both the CON and LP treatments. However, in the LB treatment, the AA content was higher compared with that in the CON and LP treatments after 7, 10, 60, and 90 days of ensiling, respectively. As expected, the concentration of AA was highest in the LB treatment after 60 and 90 days of ensiling. The PA content showed an increasing trend during the 90-day fermentation period in all three treatments, with no significant ($P > 0.05$) differences observed among the CON, LB, and LP treatments, except for the 90-day ensiling period. There were no detectable levels of BA found in the examined oat silages. However, as the ensiling process advanced, the concentrations of $NH_3$-N continued to rise in all silages. The LP treatment had lower concentrations of $NH_3$-N compared with CON and LB treatments, except for the 7-day ensiling period.

**TABLE 3**  Fermentation characteristics of ensiled oat[a]

| Item | Day | Treatment CON | LB | LP | SEM | P-value Inoculants | Fermentation | Interaction |
|---|---|---|---|---|---|---|---|---|
| pH | 7 | 4.64[aA] | 4.73[aA] | 4.42[bA] | 0.01 | <0.001 | 0.157 | 0.888 |
| | 10 | 4.55[aA] | 4.65[aA] | 4.38[bA] | | | | |
| | 60 | 4.55[abA] | 4.64[aA] | 4.38[bA] | | | | |
| | 90 | 4.55[bA] | 4.65[aA] | 4.40[cA] | | | | |
| LA (g kg$^{-1}$ DM) | 7 | 4.54[bC] | 3.60[bA] | 6.27[aA] | 0.036 | <0.001 | 0.037 | 0.035 |
| | 10 | 4.82[bB] | 4.65[bC] | 6.31[aA] | | | | |
| | 60 | 5.45[abA] | 4.54[bA] | 6.57[aA] | | | | |
| | 90 | 5.17[aAB] | 4.26[bA] | 6.61[aA] | | | | |
| AA (g kg$^{-1}$ DM) | 7 | 1.90[aB] | 2.21[aA] | 0.71[bB] | 0.11 | <0.001 | <0.001 | 0.059 |
| | 10 | 1.87[aB] | 2.45[aA] | 0.74[bB] | | | | |
| | 60 | 2.37[aAB] | 2.51[aA] | 1.38[bA] | | | | |
| | 90 | 2.42[aA] | 2.51[aA] | 1.70[bA] | | | | |
| PA (g kg$^{-1}$ DM) | 7 | 0.08[aA] | 0.12[aA] | 0.04[bB] | <0.01 | <0.001 | 0.002 | 0.712 |
| | 10 | 0.09[abA] | 0.14[aA] | 0.09[bB] | | | | |
| | 60 | 0.14[aA] | 0.16[aA] | 0.05[bB] | | | | |
| | 90 | 0.14[aA] | 0.16[aA] | 0.15[aA] | | | | |
| $NH_3$-N (g kg$^{-1}$ DM) | 7 | 1.61[aA] | 1.70[aA] | 1.65[aA] | 0.06 | 0.076 | 0.19 | 0.035 |
| | 10 | 1.83[aA] | 1.69[aA] | 1.65[aA] | | | | |
| | 60 | 2.01[aA] | 1.86[aA] | 1.73[aA] | | | | |
| | 90 | 2.13[aA] | 2.02[aA] | 1.91[aA] | | | | |

[a]Significant differences were indicated by lowercase letters (a–c) among the treatments on the same ensiling days at $P < 0.05$. Significant differences were indicated by capital letters (A–B) within the same treatment and on the same ensiling days at $P < 0.05$.DM, dry matter; LA, lactic acid; AA, acetic acid; PA, propionic acid; CON, control; LB, *Lentilactobacillus buchneri*; LP, *Lactiplantibacillus plantarum*; SEM, standard error of means. Butyric acid was not detected in all silage.

## Inoculants altered the composition and diversity of the bacterial community of the oat silage

The bacterial diversity and community compositions of oat silage were analyzed through 16S rRNA amplicon sequencing of oat materials and silage bacteria, resulting in an average of 159–285 effective tag sequences per sample. During the silage fermentation process, the alpha diversity (Shannon index) of bacterial communities in the LB and LP treatments was markedly ($P < 0.05$) lower than that in the CON treatment. Notably, the alpha diversity in the LP treatment was significantly ($P < 0.05$) lower than that in the LB treatment (Fig. 1A). The present study used principal coordinates analysis based on unweighted UniFrac distances and weighted UniFrac distances to analyze the beta diversity of oat silage bacteria and identify the factors influencing the differences between them. The results showed a clear succession of bacteria throughout the fermentation process. However, there was no distinct separation of bacterial communities observed between silages treated with or without *L. plantarum* and *L. buchneri*, as indicated in Fig. 1B and C. The changes in the bacterial community of oat silage at the phylum and genus level are presented in Fig. 1D and E, respectively. Before ensiling, the predominant phylum was *Proteobacteria*, comprising 99% of the bacterial community (Fig. 1D), while the major bacterial genera were *Erwinia* (60.99%) and *Pseudomonas* (1.80%). During the ensiling process, the bacterial community compositions of the three silage groups became similar and underwent regular changes as fermentation progressed. Throughout the fermentation process, there was a rapid increase in the relative abundance of *Lactobacillus*, while the abundance of *Erwinia* and *Curtobacterium* decreased (Fig. 1E). *Lactobacillus* dominated the fermentation process from 7 to 90 days after ensiling. However, the abundance of *Lactobacillus* in the CON treatments was lower compared with that in the LB and LP treatments.

To examine the impact of the fermentation process on bacterial taxa in silages treated with or without inoculants, a LEfSe analysis was performed, as shown in Fig. 3A through C. After fermenting oat silage for 7 days, desirable bacteria, such as *Lactococcus*, were

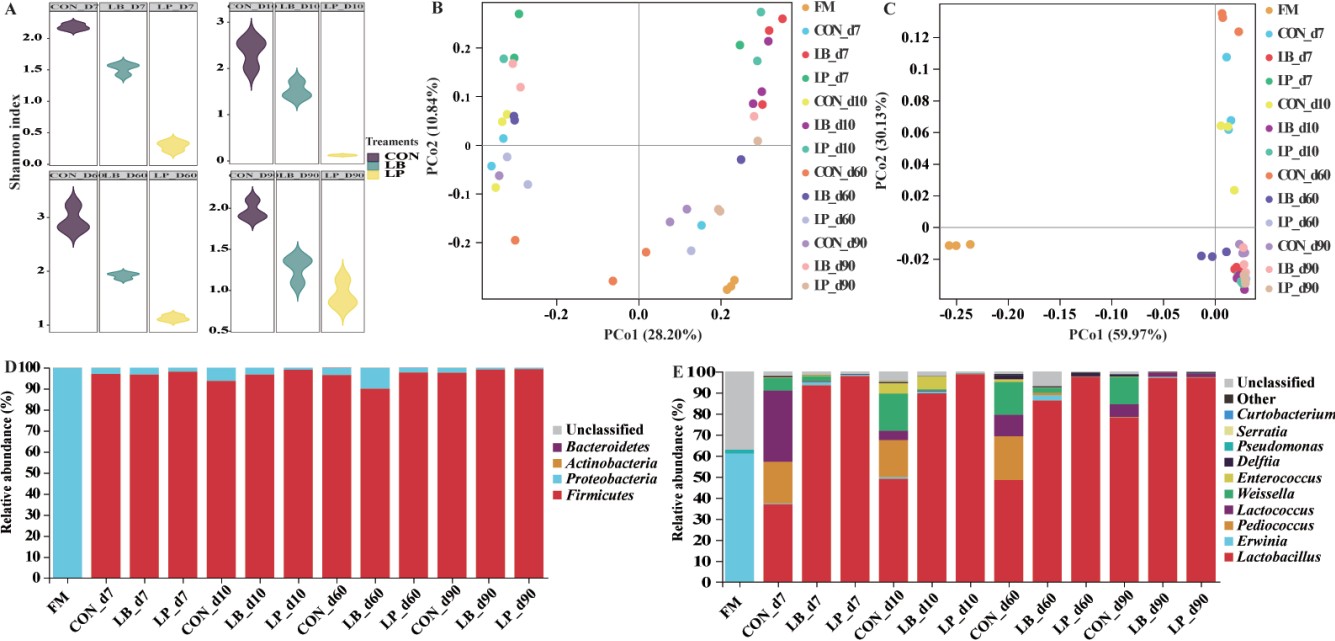

**FIG 1** Bacterial community diversities and dissimilarities of oat silage. CON, samples without inoculants; FM, fresh material; LB, samples inoculated with *Lentilactobacillus buchneri*; LP, samples inoculated with *Lactiplantibacillus plantarum*. (A) The variations of alpha-diversities of bacterial communities (Shannon index). (B and C) The bacterial community dissimilarities in different treatments and ensiling time, calculated by unweighted UniFrac and weighted UniFrac distances, with coordinates calculated by principal coordinates analysis, respectively. (D and E) Relative abundances of oat silage bacterial phylum and genus across different treatments and fermentation time, respectively.

supplemented with CON treatment. In contrast, undesirable bacteria, such as *Weissella* and *Erwinia*, were supplemented in the LB and LP treatments. After 10 days of fermentation, no significant ($P > 0.05$) differences were found in the LP treatment compared with that in the CON and LB treatments. In particular, the genus *Enterococcus* were abundant in the CON and LB treatments. After 60 days of fermentation, no significant ($P > 0.05$) differences were found in the LP treatment compared with those in the CON and LB treatments. Nevertheless, undesirable bacteria such as *Gammaproteobacteria* were enriched in the LB treatment. In oat silage fermented for 90 days, desirable bacteria such as *Lactobacillaceae* were enriched in the CON and LB treatment, and *Lactococcus* were supplemented with LP treatment.

## Inoculants altered the composition and diversity of the fungal community of oat silage

The ITS rRNA amplicon sequencing of oat materials and silage fungi produced an average of 63,752 effective tags per sample. The fungal diversity and fungal community compositions of oat silage are displayed in Fig. 2. The alpha diversity, as indicated by the Shannon index, exhibited distinct patterns. Initially, for the first 7 and 10 days of ensiling, the LB and LP treatments displayed significantly ($P < 0.05$) higher alpha diversity compared with the CON treatment, as depicted in Fig. 2A. However, as the ensiling process extended to 60 and 90 days, a notable ($P < 0.05$) decrease in alpha diversity was observed in both the LB and LP treatments. To assess the impact of various factors on the differences between oat silage fungi (beta diversity), a PCoA plot was generated. This plot utilized both unweighted and weighted UniFrac distances as the metrics for evaluating fungal diversity. The results showed a significant succession of fungi during the fermentation process, but there was no clear separation observed among the silages treated with or without *L. plantarum* and *L. buchneri* (Fig. 2B and C). The dynamics of the fungal community in oat silage at the phylum and genus level are listed in Fig. 1D and E, respectively. Before ensiling, the most representative phyla were *Basidiomycota* (55.14%), *Ascomycota* (18.10%), and *Anthophyta* (22.61%), accounting for approximately 95% (Fig.

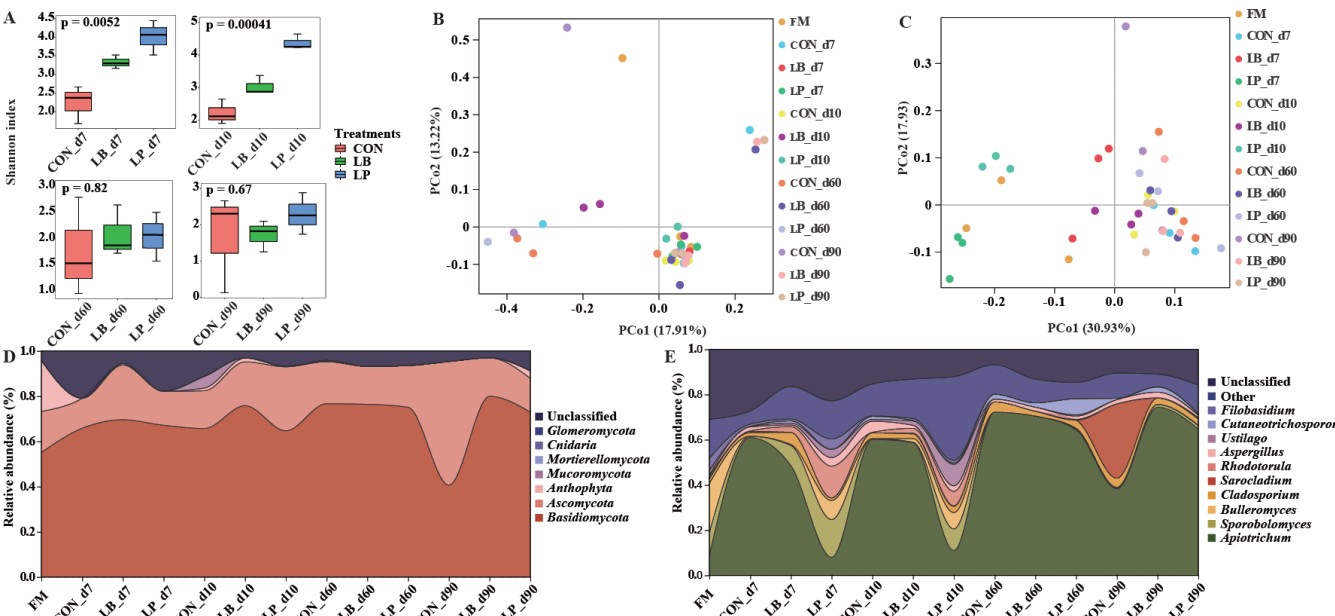

**FIG 2** Fungal community diversities and dissimilarities of oat silage. CON, samples without inoculants; FM, fresh material; LB, samples inoculated with *Lentilactobacillus buchneri*; LP, samples inoculated with *Lactiplantibacillus plantarum*. (A) The variations of fungal community alpha-diversities (Shannon index). (B and C) The fungal community dissimilarities in different treatments and ensiling time, calculated by unweighted UniFrac and weighted UniFrac distances, with coordinates calculated by principal coordinates analysis, respectively. (D and E) Relative abundances of oat silage fungal phylum and genus across different treatments and fermentation time, respectively.

2D). The main fungal genera were *Bulleromyces* (21.66%), *Sporobolomyces* (10.04%), and *Apiotrichum* (8.86%). After the ensiling process, there were noticeable regular changes in the composition of the fungal communities in all three silage groups, although the overall composition was similar. The relative abundance of *Bulleromyces*, *Sporobolomyces*, and *Apiotrichum* decreased rapidly the overall fermentation process (Fig. 2E). From day 7 to 10 after ensiling, *Apiotrichum* markedly decreased in the LP treatment versus the CON and LB treatments. In day 60 of fermentation, the dominant fungal genera found in all three silage groups were *Apiotrichum* and *Aspergillus*. However, *Aspergillus* was significantly more abundant in the CON treatment compared with the LB and LP treatments. The LEfSe analysis was performed to investigate the impact of fermentation process and inoculation treatments on fungal taxa in the oat silage samples (Fig. 3D and E). No significant (*P* > 0.05) differences were found in the overall fermentation process in the CON treatment. Following 7 days of ensiling, the fungal communities within the LB and LP treatments exhibited notable diversity. By the 10th day of fermentation, *Erythrobasidiaceae* became dominant in both the LB and LP treatments, accompanied by an enrichment of various other fungal species. After a 90-day fermentation period, *Apiotrichum* became prominent in both LB and LP treatments. However, it's worth noting that less desirable fungi, such as *Aspergillus*, were more prevalent in the LB treatment.

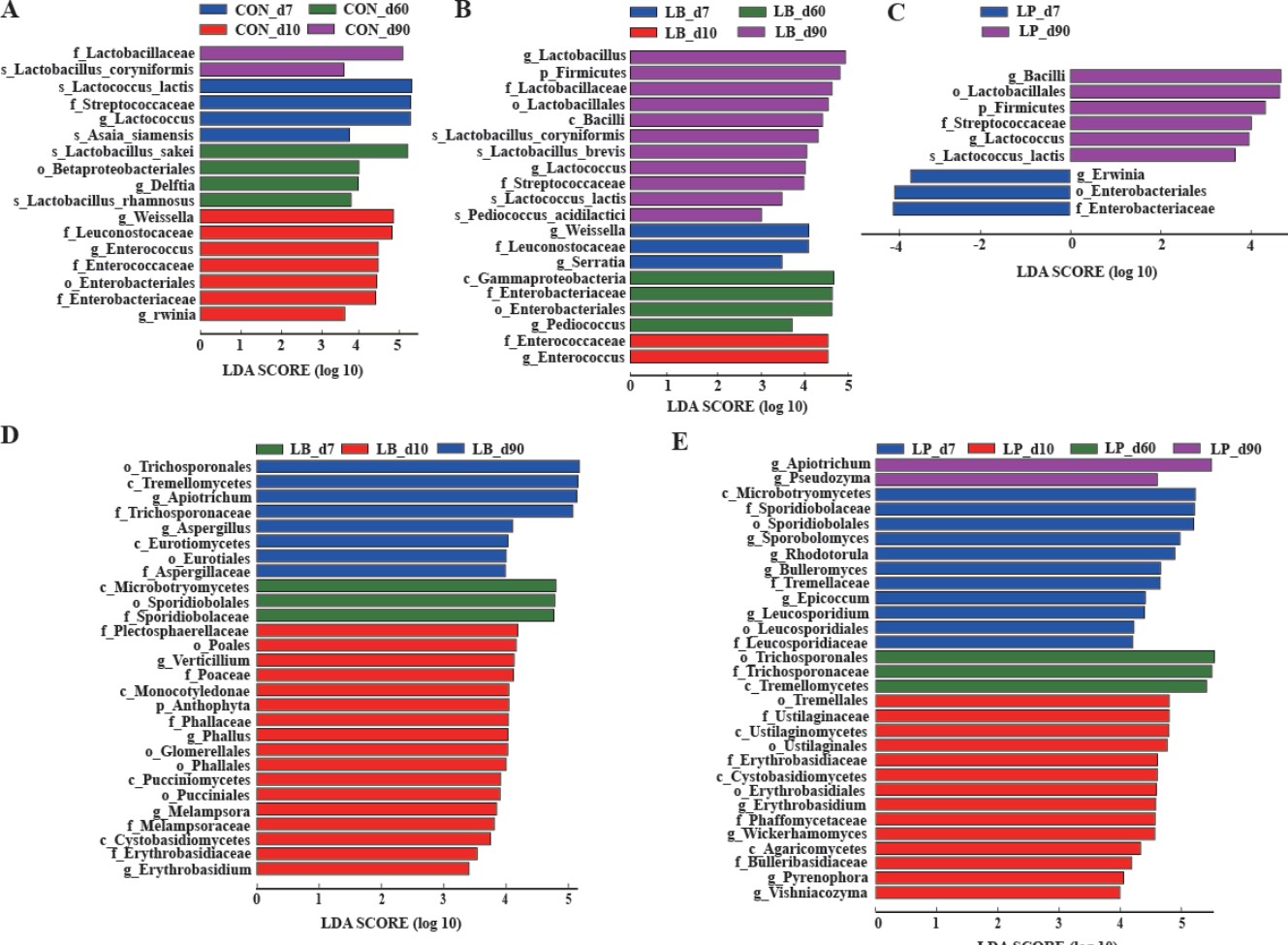

**FIG 3** Differences of microbial taxa in whole oat silage with different treatments. CON, samples without inoculants; LB, samples inoculated with *Lentilactobacillus buchneri*; LP, samples inoculated with *Lactiplantibacillus plantarum*. The LEfSe analysis of oat silage bacterial (A–C) and fungal (D, E) biomarkers associated with inoculants for different fermentation time. (A) Bacteria of samples without inoculants. (B) Bacteria of samples inoculated with *Lentilactobacillus buchneri*. (C) Bacteria of samples inoculated with *Lactiplantibacillus plantarum*. (D) Fungi of samples inoculated with *Lentilactobacillus buchneri*. (E) Fungi of samples inoculated with *Lactiplantibacillus plantarum*.

## Associations between the microbiome, chemical compositions, and fermentation quality

We used a heatmap to assess the correlations between the top 10 bacterial genera and the chemical constituents, fermentation quality, and fertilization profile. This analysis was conducted using Spearman's correlation coefficient (Fig. 4A and B). As shown in Fig. 4A, the DM and lignin contents had a significantly negative correlation with *Lactobacillus* (DM: rho = −0.508, *P* < 0.01; lignin: rho = −0.548, *P* < 0.01). The contents of ADF and MDF also had a significant negative correlation with *Lactobacillus* (ADF: rho = −0.669, *P* < 0.01; NDF: rho = −0.769, *P* < 0.01). Notably, several other significant correlations emerged within the data set. Specifically, both ADF and NDF showed a strong positive correlation with *Enterococcus* (ADF: rho = 0.622, *P* < 0.01; NDF: rho = 0.710, *P* < 0.01) and *Weissella* (ADF: rho = 0.691, *P* < 0.01; NDF: rho = 0.608, *P* < 0.01). Additionally, a substantial positive correlation was observed between CP and *Lactobacillus* (rho = 0.754, *P* < 0.01), while a significant negative correlation was found between CP and *Enterococcus* (rho = −0.719, *P* < 0.01), *Weissella* (rho = −0.707, *P* < 0.01), *Lactococcus* (rho = −0.554, *P* < 0.01), and *Pediococcus* (rho = −0.479, *P* < 0.01). Furthermore, the WSC content exhibited noteworthy negative correlations with *Weissella* (rho = −0.552, *P* < 0.01) and *Lactococcus* (rho = −0.369, *P* = 0.02). These correlations between bacterial genera and various fermentation characteristics are visually represented in Fig. 4B. In addition, LA concentration displayed significantly positive correlations with *Lactobacillus* (rho = 0.457, *P* < 0.01) and negative

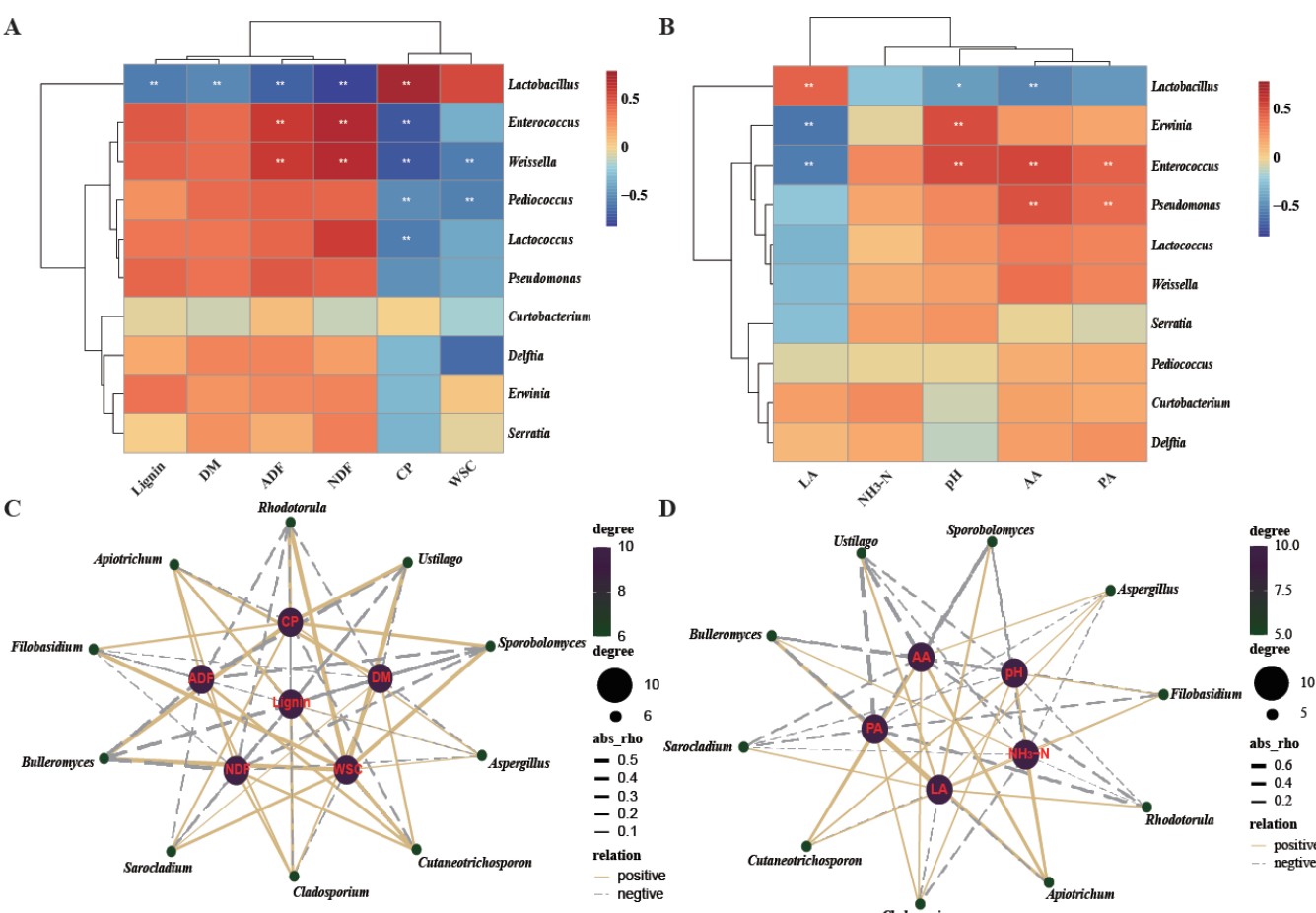

**FIG 4** Correlation analysis between the genera (top 10 significant genera) and the fermentation characteristics of oat silage. (A) Correlation analysis between bacteria and chemical compositions. (B) Correlation analysis between bacteria and fermentation quality. (C) Correlation analysis between fungi and chemical compositions. (D) Correlation analysis between fungi and fermentation quality. *Significant correlation at the *P* < 0.05 level. **Significant correlation at the *P* < 0.01 level.

correlations with *Erwinia* (rho = −0.587, *P* < 0.01) and *Enterococcus* (rho = −0.546, *P* < 0.01). The bacterial communities were not significantly (*P* > 0.05) affected by the $NH_3$-N content. The study revealed a significant negative correlation between pH and *Lactobacillus* (rho = −0.418, *P* < 0.01) and a positive correlation with *Erwinia* (rho = 0.530, *P* < 0.01) and *Enterococcus* (rho = 0.548, *P* < 0.01). Furthermore, as anticipated, there was a significant negative correlation between the concentration of AA and *Lactobacillus* (rho = −0.5245, *P* < 0.01). Based on Spearman analysis, the network evaluated the correlations between the top 10 fungal genera and chemical constituents/fermentation profile (Fig. 4C and D). The results showed a significant negative correlation between *Aspergillus* and the DM, ADF, and NDF contents (DM: rho = −0.204, *P* = 0.204, ADF: rho = −0.036, *P* = 0.837; NDF: rho = −0.141, *P* = 0.412). Additionally, LA showed a significant positive correlation with *Bulleromyces* (rho = 0.400, *P* = 0.016), *Ustilago* (rho = 0.236, *P* = 0.167), *Sporobolomyces* (rho = 0.217, *P* = 0.205), *Aspergillus* (rho = 0.033, *P* = 0.845), and *Filobasidium* (rho = 0.001, *P* = 0.993). As expected, the pH had a significantly negative correlation with these fungi.

## DISCUSSION

Silage is an effective technique to extend the availability of feed for ruminants (36). Globally, the LAB inoculants, including *L. buchneri* and *L. plantarum*, are commonly employed in the process. Through anaerobic fermentation, the chemical constituents of forages and grass can be well maintained while microorganisms generate organic acids, mainly LA and AA, which can enhance the nutritional profile of the silage (3). Unfortunately, there is limited information available regarding the dynamics of oats inoculated with or without *L. buchneri* or *L. plantarum*. In this study, a combination of multiple physicochemical analyses and high-throughput sequencing was employed to investigate the dynamics of fermentation quality and microbial community of oat silage during anaerobic fermentation with or without inoculation of *L. buchneri* or *L. plantarum*. This is the first study that integrates bacteria and fungi in response to anaerobic bioaugmentation of oat ensiling with *L. buchneri* or *L. plantarum*, providing a tentative model for future research in this area.

### Chemical and microbial content of oat

In this study, the ADF and NDF contents were found to be in line with those reported by Wang et al. (37). However, it is worth noting that the present CP and WSC contents were notably higher. These differences can be attributed to various factors such as plant species, genotype, sowing density, fertilization, irrigation, harvest time, and environmental conditions (37). The WSC content in forages and grasses plays a pivotal role in facilitating LAB fermentation. As stipulated by Wang et al. (38), good quality silages should ideally have a WSC content exceeding 5% of DM. In our study, the WSC content, which stands at 16.30%, comfortably surpasses this requirement. However, the fermentation process is also dependent on the number of LAB present, and the minimum requirement for the number of LAB in fresh materials should be greater than 5.0 log cfu/g FW according to You et al. (22). Despite meeting the WSC content requirements, the presence of lower LAB counts (<5.0 log cfu/g of FW) may lead to undesirable fermentation and end products (39). Hence, it is important to examine the role of LAB in influencing silage fermentation and the associated changes in the microbiome.

### Fermentation aspects and chemical content of oat for silage

Compared with raw materials, the WSC content decrease in the first 7 days of the fermentation process is critical to inhibit undesirable microorganisms and reduce nutrition loss (7, 40). It is interesting to note that there was a decrease in pH and an increase in LA and AA concentrations during the first 7 days of ensiling in all treatments. The pH level serves as a crucial parameter when evaluating the quality of fermentation, and its decline is attributed to the acidification resulting from the production of LA

and AA by LAB from WSC (22, 40). As the fermentation process prolonged, the WSC content decreased and the pH, LA, and AA concentrations increased. Additionally, the LP treatment showed lower pH and higher LA concentration compared with the CON and LB treatments, while the LB group exhibited higher AA and PA concentrations than the CON and LP treatments. These differences may be attributed to the specific fermentation characteristics of *L. buchneri* and *L. plantarum*. Previously published reports indicated that heterofermentative LAB, for example, *L. buchneri*, could improve aerobic stability by increasing the content of AA and PA via the pentosephosphate pathway (Fig. 5) (7, 41). Furthermore, homofermentative LAB, such as *L. plantarum*, are capable of producing LA from glucose, pentoses, and 3xylose through various metabolic pathways, including the Emden–Meyerhoff pathway, the phosphoketolase pathway, and pentosephosphate pathway (Fig. 5) (5, 42, 43). Therefore, the higher pH was observed in the LB treatment than that in the CON and LP treatments, because the LA was broken down into AA and PA by the *L. buchneri*. Similarly, the LP treatment had a higher CP content and lower ADF, NDF, lignin, and $NH_3$-N contents compared with the CON and LB treatments. However,

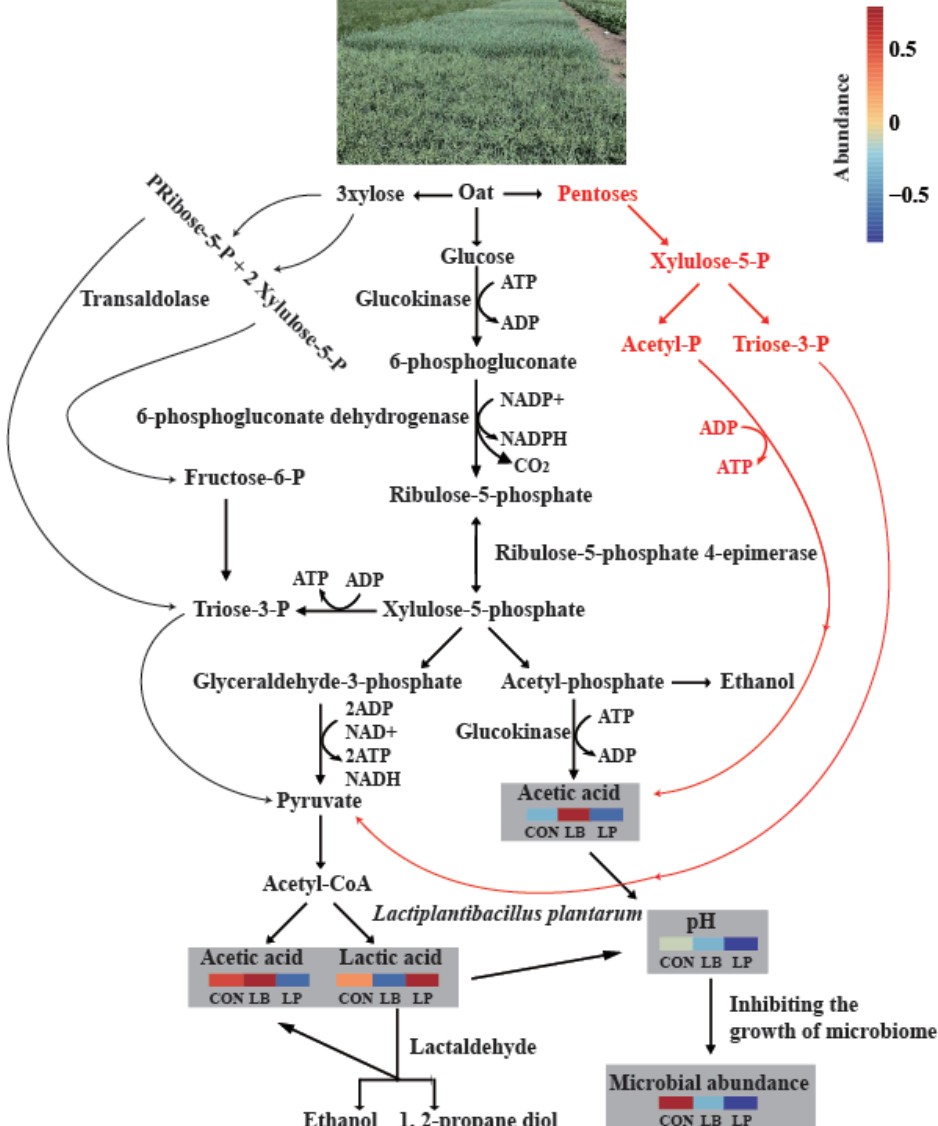

**FIG 5** Overview on LAB. The black indicated the homofermentative metabolism of *Lactiplantibacillus plantarum* by the Emden–Meyerhoff pathway, phosphoketolase pathway, and pentosephosphate pathway. The red indicated heterofermentative metabolism of *Lentilactobacillus buchneri* via the pentosephosphate pathway (5, 41–43).

protein degradation is a common occurrence during the fermentation process due to the action of plant enzymes and microbial metabolism (40). The lower CP content and higher NH$_3$-N content in the current study suggest that protein degradation occurred during ensiling. This degradation is likely attributed to a combination of plant enzymes and microbial metabolism. This was particularly evident in the CON and LB treatments, where the NH$_3$-N content was higher. The lower pH in the LP treatment may have inhibited the growth and metabolism of undesirable microorganisms such as *Clostridium* and *Aspergillus*, leading to less protein degradation, which is consistent with previous studies (22, 44). Additionally, during the ensiling process, the digestible cell wall is broken down by hydrolytic activities, including microbial activities, enzymatic, and acidolysis (6, 22).

## Bacterial variety in oat silage

During the fermentation process of oat silage, the bacterial diversity and composition were analyzed using 16S rRNA sequencing, allowing for a comprehensive understanding of changes throughout the process. Alpha diversity was used to measure the variance of the bacterial community. The coverage values of all samples exceeded 0.99 (data not shown), indicating that the depth of sequencing was sufficient to accurately reflect the bacterial community (6). The bacterial diversity of oat silage was analyzed using the Shannon index in this study, and significant differences were observed with the prolongation of the fermentation process. These findings are consistent with previous studies that have reported decreased alpha diversity due to the inhibition of undesirable microorganisms by pH, which are gradually replaced by LAB (6, 7). The PCoA plot clearly showed the variance of the bacterial community through the separation of the three treatments based on the fermentation period. This indicated that the fermentation process and the use of additives had notable effects on the bacterial community of oat silage. The bacterial phylum *Proteobacteria* was found to be the predominant phylum in the FM, while the phylum *Firmicutes* was identified as the most dominant phylum throughout all fermentation periods. Previous studies found that *Proteobacteria* is comprised of the genera *Pantoea*, *Pseudomonas*, *Sphingomonas*, and *Erwinia* in silage (40, 45). The current findings are consistent with prior research indicating a transition from *Proteobacteria* to *Firmicutes* as the dominant phylum during the fermentation of oat silage. This shift is likely due to Firmicutes microorganisms thriving under low pH and anaerobic conditions, as well as the decrease in WSC after ensiling. These factors contribute to the change in bacterial community composition during fermentation (40, 46, 47). There was a significant decrease in the abundance of *Erwinia* during ensiling. This observation aligns with previous research findings that indicate *Erwinia* tends to be inhibited in anaerobic fermentation under acidic conditions (pH < 5.40) (48, 49). Within the first 7 days of ensiling, *Lactobacillus* was the dominant bacterial genus in the LB and LP treatments, accounting for more than 90% of the bacterial community. The addition of *L. buchneri* and *L. plantarum* may have accelerated the fermentation process and improved the growth of *Lactobacillus*, which may be the main reason for this dominance.

In contrast, within the CON treatment, there was a decrease in the abundance of *Lactococcus* and *Pediococcus*, while the abundance of *Lactobacillus* increased as the fermentation period extended. Previous studies have shown that *Lactococcus* and *Pediococcus* initiate fermentation at the early stage of ensiling and are eventually replaced by *Lactobacillus* with more acid-tolerant characteristics as the pH decreases over time (50). Therefore, the higher level of *Lactobacillus* than those of *Lactococcus* and *Pediococcus* was observed as the ensiling time extended.

## Fungal diversity of oat silage

In general, fungal studies have predominantly concentrated on aerobic spoilage in silages. However, the progression of the fungal community during fermentation, particularly in oat silage, remains an area that has not been comprehensively elucidated. Here, we analyzed the fungal community of oat silage was analyzed using ITS

amplicon sequencing. Good's coverage values of all samples were above 0.99 (data not shown), which is similar to the bacterial community results, indicating that the sequencing depth was sufficient for reliable analysis (51). The alpha diversity of the fungal community was assessed by calculating the Shannon index. This index served as a reflection of the diversity within the fungal community and was used for comparative purposes among the three groups. The LB and LP treatments exhibited an initial increasing trend in the first 7 and 10 days of ensiling, followed by a subsequent decrease in diversity during the 60 and 90 days of ensiling. This pattern differed from the trend observed in bacterial diversity. The reason for this difference could be that the initial aerobic conditions of ensiling favored fungal growth over bacterial growth, resulting in a higher rate of fungal growth at the early stage. However, as the ensiling process progressed, the consumption of oxygen by aerobic microorganisms created an anaerobic and acidic environment that was unsuitable for fungal growth, leading to a reduction in fungal diversity (51, 52). In the FM, the primary fungal phylum identified was *Basidiomycota*, followed by *Ascomycota* and *Anthophyta*. Interestingly, the dominant fungal phyla, *Basidiomycota* and *Ascomycota*, remained consistent throughout all stages of the fermentation period. These findings differ from those of a prior study, which reported *Chytridiomycota*, *Ascomycota*, *Zygomycota*, and *Basidiomycota* as the dominant fungal phyla in elephant grass in the subtropical zone (53). Additionally, the high relative abundance of *Apiotrichum*, *Sporobolomyces*, and *Bulleromyces* in the FM was inconsistent with that of the study by Grazia et al. who found that *Geotrichum* was the dominant fungi in maize materials (54). It is possible that these discrepancies are due to differences in the materials and environmental conditions used in the studies. After ensiling, the abundance of *Bulleromyces* decreased, possibly due to its enrichment in postharvest samples (55). During the first 7 and 10 days of ensiling, the dominant genera in the LP group were different from those in the CON and LB treatments. After 90 days of ensiling, *Apiotrichum* and *Sarocladium* became the primary genera, which differed from the LB and LP groups where *Apiotrichum* and *Aspergillus* were dominant.

These abovementioned findings could be attributed to the catabolic reactions that provide necessary nutrients for the growth of these fungal genera (52). However, these results are not entirely consistent with previous work on dominant fungal phyla in other ensiled materials such as elephant grass, fermented defatted rice bran, and sainfoin silage (36, 52, 56). The differences observed here may be attributed to the sampling time and the stage of plant growth. *Aspergillus* species are widely used in fermentation processes to enhance nutritional value due to their ability to produce various metabolites (52, 57). The genus Aspergillus is known for its ability to produce various enzymes including cellulase, hemicellulase, and xylanase (52), which can contribute to the breakdown of plant cell wall components. As a result, the LP treatment had significantly lower ADF, NDF, and lignin contents compared with the other groups.

## Correlation analysis of microbial community and fermentation characteristics in oat silage

Network analysis has become a popular tool for investigating the relationships between a set of nodes and items (58). In microbiome research, correlation networks provide a visual summary of the abundance of information and can help identify potential correlations between silage microbiome and silage quality (38). The diversity of the microbial community can reflect variations in physicochemical characteristics, and the use of additives and fermentation period can significantly affect silage quality and microbial community (38, 39, 59). By studying the correlations between the microbial community and silage characteristics, we can gain a better understanding of the key bacteria and fungi that contribute to silage quality (60). In the current study, a positive correlation was found between *Lactobacillus* and CP and LA, whereas a negative correlation was observed with pH and AA, which is consistent with previous research (60). These findings suggest that *Lactobacillus* may have a significant role in improving fermentation quality and preserving oat silage. Additionally, *Enterobacteriaceae* was

positively correlated with pH, LA, and AA but negatively correlated with LA. It should be noted that Enterococcus, a cocci LAB genus, can only thrive in a pH > 4.5 environment (48). However, *Enterobacteriaceae* could transfer LA to AA and PA in silage (59). In this study, *Pseudomonas* was found to be positively associated with AA and PA. This could be explained by the fact that *Pseudomonas* converts D-galacturonate to pyruvate and produces AA and other products. The fungal community also plays an important role in the development of physical characteristics in silages. However, the correlation between fungal community and fermentation characteristics was found to be limited in silage. The genus *Sporobolomyces*, often found in low-nutrition environments, exhibited positive associations with CP and WSC. Conversely, it displayed negative associations with ADF, NDF, lignin, AA, and PA. These trends were consistent with findings from a prior study (61). The prevalence of the genus *Sporobolomyces* in low-nutrition environments could provide an explanation for these observed results (62).

However, there is currently limited information on the fungal community and its role in silage functions, despite the use of amplicon sequencing. Furthermore, the specific roles of *Sporobolomyces* in silage are not yet fully understood (61, 62). High-throughput sequencing was employed to investigate the alterations in the microbial community, structure, and functions, encompassing both bacteria and fungi, throughout the fermentation of oat silage. The results revealed the activation of numerous metabolic pathways throughout the fermentation process, with various bacteria and fungi being linked to these pathways. It could be extracted that the microbiome demonstrated adaptability and responsiveness to the evolving environment. These findings are in line with prior research in microbial ecology and fermentation. Notably, *L. buchneri* and *L. plantarum* emerged as pivotal contributors to anaerobic bioaugmentation during oat silage fermentation.

## Conclusions

In conclusion, this study investigated the effect of *L. buchneri* or *L. plantarum*-mediated anaerobic bioaugmentation on the fermentation quality, microbial community, and metabolic function of oat silage. The results indicated that both LAB strains improved the ensiling performance of oat silage, with *L. plantarum* showing more pronounced effects on decreasing pH and $NH_3$-N and increasing LA concentration. This study provides insights into the potential application of LAB-mediated bioaugmentation in improving the quality of oat silage and highlights the importance of microbial community regulation in ensiling.

### ACKNOWLEDGMENTS

The authors acknowledge the financial support provided by several sources, including the Selection and Mechanism of Low Temperature Resistant Lactic Acid Bacteria in Natural Forage Silage in Hulunbuir (NJZY21238), Key Cultivation Technology and Development Program of High Yield and Quality in Hulunbuir Area (GH202001), Grassland and Grass Industry Scientific Data Resource Construction and Sharing Service (2022–22), and China Agriculture Research System (CARS-34).

### AUTHOR AFFILIATIONS

[1]College of Agriculture and Forestry, Hulunbuir University, Hulunber, China
[2]Institute of Agricultural Resources and Regional Planning, Chinese Academy of Agricultural Science, Hulunber Grassland Ecosystem Observation and Research Station, Beijing, China
[3]Inner Mongolia Academy of Agricultural Science & Animal Husbandry, Hohhot, China
[4]Key Laboratory of Forage Cultivation, Processing and High Efficient Utilization, College of Grassland, Resources and Environment, Inner Mongolia Agricultural University, Hohhot, China

## AUTHOR ORCIDs

Lijun Xu  http://orcid.org/0000-0003-2507-6557
Shuai Du  http://orcid.org/0000-0002-2035-0927

## AUTHOR CONTRIBUTIONS

Yanzi Xiao, Conceptualization, Formal analysis, Funding acquisition, Investigation, Methodology, Visualization, Writing – original draft | Lin Sun, Conceptualization, Formal analysis, Investigation | Xiaoping Xin, Supervision, Writing – review and editing | Lijun Xu, Funding acquisition, Supervision, Writing – review and editing | Shuai Du, Conceptualization, Data curation, Investigation, Methodology, Validation, Writing – original draft, Writing – review and editing

## DATA AVAILABILITY

The raw data sequencing files and associated metadata of these samples have been uploaded into the NCBI's Sequence Read Archive with the accession number PRJNA863426.

## ADDITIONAL FILES

The following material is available online.

Open Peer Review

PEER REVIEW HISTORY (review-history.pdf). An accounting of the reviewer comments and feedback.

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
