## [Reviewer comments · Microbiology Spectrum]

Microbiology Spectrum

Physicochemical characteristics and microbial community succession during oat silage prepared without or with *Lactiplantibacillus plantarum* or *Lentilactobacillus buchneri*

Yanzi Xiao, Lin Sun, Xiaoping Xin, Lijun Xu, and Shuai Du

Corresponding Author(s): Shuai Du, Zhejiang University

Review Timeline:

Submission Date:	May 26, 2023
Editorial Decision:	August 7, 2023
Revision Received:	September 17, 2023
Accepted:	October 7, 2023

Editor: Jing Han

Reviewer(s): Disclosure of reviewer identity is with reference to reviewer comments included in decision letter(s). The following individuals involved in review of your submission have agreed to reveal their identity: Changrong Wu (Reviewer #2)

Transaction Report:

DOI: <https://doi.org/10.1128/spectrum.02228-23>

August 7, 2023

Dr. Shuai Du
Zhejiang University
Yuhangtang Road 866
Hangzhou
China

Re: Spectrum02228-23 (Physicochemical characteristics and microbial community succession during oat silage prepared without or with *Lactiplantibacillus plantarum* or *Lentilactobacillus buchneri*)

Dear Dr. Shuai Du:

Link Not Available

Sincerely,

Jing Han

Journals Department
Reviewer comments:

Reviewer #1 (Comments for the Author):

This study determined the physicochemical characteristics and microbial community succession during oat silage prepared without or with *Lactiplantibacillus plantarum* or *Lentilactobacillus buchneri* by using 16S and ITS amplicon sequencing methods. This study is well done! The results found that the addition of LAB regulated the microbial community in oat silage, which in turn influenced the ensiling products. The presented results are not only provide the potential application of LAB-mediated bioaugmentation in improving the quality of oat silage but also highlight the importance of microbial community regulation in ensiling. The layout of the paper is suitable. The given methodology is well detailed. The discussion is clear and easy to follow. The article should be of interest for the readers but before publication a depth revision should be addressed. The authors should

pay much more attention in preparing the manuscript. It is recommended to polish the language, and some sentences are not smooth, such as the first sentence of second paragraph in 3.5. Overall, this ms can be accepted after a minor revision.

Specific suggestions are as follows:

It is suggested to introduce the materials and methods in the background of the abstract, mainly including the treatment.

Line 25: Please check the name of *Lactiplantibacillus plantarum* or *Lentilactobacillus buchneri*, *Lactobacillus plantarum* or *Lactobacillus buchneri*.

Line 27: A lot of information is missing in the abstract section. Readers can not get enough information about the methods from this section.

Line 30: can → could

Line 33: significantly ($p < 0.05$)

Line 35 - 36: From 7 to 90 days of ensiling, *Lactobacillus* was the dominant genus during the whole fermentation process. Is that all groups? Please specify.

Line 38: The homofermentative *L. plantarum* or heterofermentative *L. buchneri* how to regulate the ensiling performance, bacterial and fungal community compositions. By enhanced or inhibited what metabolic pathways.

Line 39 and 41: First-time appearance of LAB and LA should give the full name.

Line 38-42: This sentence should be rewrite. It is suggested to rewrite the last sentence of the conclusion because it is too detailed and should summarize the main conclusions.

Line 44: The authors should refer to the submission guidelines to correct the format of key words.

Line 48: '*Avena sativa*' should be italic.

Line 74: Please list the full name of the LA when it first appear.

Line 76: ", , "?".

Line 78: The lactic acid (LA) should be use an abbreviation.

Line 91 and elsewhere: '*L. plantarum*' should be italic.

Line 97: In the materials and methods. It is suggested that the test procedure should be rewritten in the order in which the tests were carried out.

Line 101: What did he mean?

Line 125: There should be a space between "AOAC, 2005" and "The".

Line 129: There's a space missing here. And '10 g' is better. The "deionized water" is the "sterile water"

Line 132 and elsewhere: "NH₃-N" not "NH₃-N"

Line 133: the ammonia nitrogen (NH₃-N) content was determined using a previously published method, please specify what method? There's a space missing here.

Line 133 - 138: Provide a short description of the determination of organic acids. Also, a description of the assessment of the microbial population is necessary.

Line 150: The polymerase chain reaction (PCR) amplification how to performed?

Line 165: There's a space missing here.

Line 184 - 185: Show how the difference between treatments was identified. E.g. a ANOVA.

Line 192: In the results, the description of some of the results analyzed in the manuscript is not very accurate and needs to be narrated in strict accordance with the results of significance. It is recommended that in the results analysis, all test treatments are represented by abbreviated symbols, which is more concise and clear.

Line 200 and elsewhere: There should be a ($p > 0.05$) here.

Line 260: The "*Erwinia*" and "*Pseudomonas*" should be italic.

Line 273-277: Please describe the object of comparison.

Line 287 and elsewhere: Please standardise the header format throughout the text. "Fig. 2." or "Figure 2".

Line 308 and elsewhere: The "*Aspergillus*" should be italic.

Line 314-315 This sentence should be rewrite.

Line 332: NDF: $\rho = 0.607777572249648$, $p < 0.01$). Please retain three decimals.

Line 344 and elsewhere: The "*Lactobacillus*" and "p" should be italic.

Line 366: What dynamics of oats.

Line 403: can → could.

Line 407: Please insert a comma before "and".

Line 537: I do not understand the link of this with the oat silage.

Line 541: "AAAA" ?

Table 2 and 3: There are some significant interaction that are not described in this section.

Figures 1 -2: which can not be read: the letters are too small

Suggest one reference to cite and check:

Using PICRUSt2 to explore the functional potential of bacterial community in alfalfa silage harvested at different growth stages

Reviewer #2 (Comments for the Author):

1. The description of microbial counts of fresh oat was not in line with Table 1.
2. Why the DM content of oats was so high? If you wilted them, the wilt condition need to be illustrated.
3. There were conflicts between 'There were no significant differences in the ADF concentration throughout all treatments.' and

'The ADF concentration was significantly decreased in L. plantarum-treated oat silages compared to both the CON and L. plantarum treatment.'

4. The reason that L. buchneri had a higher pH compared to the CON should be explained.

5. The resolution of figures used in the manuscript too low to be clear reviewed.

6. Line 384 was wrong.

Staff Comments:

Preparing Revision Guidelines

Please return the manuscript within 60 days; if you cannot complete the modification within this time period, please contact me. If you do not wish to modify the manuscript and prefer to submit it to another journal, please notify me of your decision immediately so that the manuscript may be formally withdrawn from consideration by Microbiology Spectrum.

1. The description of microbial counts of fresh oat was not in line with Table 1.
2. Why the DM content of oats was so high? If you wilted them, the wilt condition need to be illustrated.
3. There were conflicts between 'There were no significant differences in the ADF concentration throughout all treatments.' and 'The ADF concentration was significantly decreased in *L. plantarum*-treated oat silages compared to both the CON and *L. plantarum* treatment.'
4. The reason that *L. buchneri* had a higher pH compared to the CON should be explained.
5. The resolution of figures used in the manuscript too low to be clear reviewed.
6. Line 384 was wrong.

Dear Editor and Reviewers

Thank you very much for evaluating our paper.

Thank you for your letter and for the reviewers' comments concerning our manuscript entitled "Physicochemical characteristics and microbial community succession during oat silage prepared without or with *Lactiplantibacillus plantarum* or *Lentilactobacillus buchneri*" (No. Spectrum 02228-23). We will be happy to edit the text further, based on helpful comments from editor and reviewers. We appreciate the editor very much for their positive and constructive comments and suggestions. According to the comments, this revised manuscript was checked by native speakers of English for editing English grammar. The comments and suggestions are not only helpful for us to revise and improve our manuscript, but also benefit our further research. We hope that our paper much better quality than before.

Best regards,

Corresponding author:

Dr. Shuai Du

E-mail: dushuai_nm@sina.com

Reviewer #1

1. It is suggested to introduce the materials and methods in the background of the abstract, mainly including the treatment.

Response: Yes, we have revised clearly as:

「The oat was harvested at the milk stage of maturity and was chopped into 30 mm size. Then, the oat was treated with the distilled water (control, CON treatment); *L. buchneri*; and *L. plantarum*, the addition of *L. buchneri* and *L. plantarum* was in 1×10^6 colony-forming units/g of fresh matter and stored at room temperature (25 °C).」

2. Line 26: Please check the name of *Lactiplantibacillus plantarum* or *Lentilactobacillus buchneri*, *Lactobacillus plantarum* or *Lactobacillus buchneri*.

Response: Yes, we have checked and revised throughout the manuscript.

3. Line 25-31: A lot of information is missing in the abstract section. Readers can not get enough information about the methods from this section.

Response: Yes, we have revised clearly as:

「This study aimed to investigate the effects of *Lactiplantibacillus plantarum* (*L. plantarum*) or *Lentilactobacillus buchneri* (*L. buchneri*) on the fermentation characteristics and microbial community of oat silages in response to additives. The oat was harvested at the milk stage of maturity and was chopped into 30 mm size. Then, the oat was treated with the distilled water (control, CON treatment); *L. buchneri*; and *L. plantarum*, the addition of *L. buchneri* and *L. plantarum* was in 1×10^6 colony-forming units/g of fresh matter and stored at room temperature (25 °C).」

4. Line 35: can→could

Response: Yes, we have revised.

5. Line 38: significantly ($p < 0.05$)

Response: Yes, we have revised.

6. Line 40 - 43: From 7 to 90 days of ensiling, Lactobacillus was the dominate genus during the whole fermentation process. Is that all groups? Please specify.

Response: Yes, we have revised clearly as:

「From 7 to 90 days of ensiling, *Lactobacillus* was the dominate genus during the whole fermentation process in the three treatments.」

7. Line 43-49: The homofermentative *L. plantarum* or heterofermentative *L. buchneri* how to regulate the ensiling performance, bacterial and fungal community compositions. By enhanced or inhibited what metabolic pathways.

Response: Yes, we have revised in the manuscript.

「The homofermentative *L. plantarum* regulated the fermentation quality and microbial community by enhanced the Emden–Meyerhoff pathway, phosphoketolase pathway and pentosephosphate pathway, and the heterofermentative *L. buchneri* modulated the ensiling performance and microbial community via improved the pentosephosphate pathway.」

8. Line 102 and 115: First-time appearance of LAB and LA should give the full name.

Response: Yes, we have revised.

9. Line 49-56: This sentence should be rewrite. It is suggested to rewrite the last sentence of the conclusion because it is too detailed and should summarize the main conclusions.

Response: Yes, we have revised.

「These results suggested that the addition of lactic acid bacteria could improve the ensiling performance by regulating the microbial community in oat silage, and *L. plantarum* was more beneficial than *L. buchneri* for enhancing the fermentation quality.」

10. Line 58: The authors should refer to the submission guidelines to correct the format of key words.

Response: Yes, we have revised.

11. Line 91: 'Avena sativa' should be italic.

Response: Yes, we have revised.

12. Line 115: Please list the full name of the LA when it first appear.

Response: Yes, we have revised.

13. Line 117: ", ,"?.

Response: Yes, we have revised.

14. Line 118: The lactic acid (LA) should be use an abbreviation.

Response: Yes, we have revised.

15. Line 130 and elsewhere: '*L. plantarum*' should be italic.

Response: Yes, we have revised throughout the manuscript.

16. Line 157: In the materials and methods. It is suggested that the test procedure should be rewritten in the order in which the tests were carried out.

Response: Yes, we have revised clearly as:

[For sampling of chemical composition parameters, clean containers were used to collect fresh materials (FM) and oat silage after being uniformly blended. The dry matter (DM) content was determined by drying a sub-sample in an oven for 72 hours at 65°C and then grinding it through a 1 mm screen (FW100, Taisite Instrument Co., Ltd., Tianjin, China) for further chemical analysis. The ANKOM A200i Fiber Analyzer

(ANKOM Technology, Macedon, NY, USA) was utilized to determine the fiber compositions, including the lignin, neutral detergent fiber (NDF), and acid detergent fiber (ADF) contents, following previous reports (Van Soest et al., 1991; Li et al., 2020). The crude protein (CP) content was determined using the method of the Association of Official Analytical Chemists (AOAC, 2005). The anthrone method (Thomas, 1977) was utilized to determine the water-soluble carbohydrate (WSC) content. 10 grams of silage samples were mixed with 90 mL sterile water and stored at 4 °C fridge for 24 h for the extractions. After extraction, the samples were filtered through four layers of cheesecloth. The pH of the filtrate was measured using a glass-electrode pH meter. The concentrations of organic acids (lactic acid (LA), acetic acid (AA), propionic acid (PA), and butyric acid (BA)) in the oat silage were determined by the high-performance liquid chromatography according to the previously reported method (You et al., 2021). The ammonia nitrogen (NH₃-N) content was determined using the phenol–hypochlorite method based on the previously published method (Broderick & Kang, 1980). The microbial populations, including LAB, yeasts, and molds, in the FM were counted using the plate count method and expressed as colony-forming units (cfu)/g of FM (You et al., 2021). The de Man, Rogosa, Sharpe agar (Difco Laboratories, Detroit, MI, USA) was used to count the numbers of LAB after incubating at 30°C for 48 h. The potato dextrose agar (Nissui

Ltd., Tokyo, Japan) was used to count the numbers of molds and yeast after incubating at 30°C for 48 h.」

17. Line 140: What did he mean?

Response: Yes, we have revised.

18. Line 165: There should be a space between "AOAC, 2005" and "The".

Response: Yes, we have revised.

19. Line 168: There's a space missing here. And '10 g' is better. The "deionized water" is the "sterile water"

Response: Yes, we have revised.

20. Line 174 and elsewhere: "NH₃-N" not "NH3-N"

Response: Yes, we have revised throughout the manuscript.

21. Line 170-176: the ammonia nitrogen (NH₃-N) content was determined using a previously published method, please specify what method? There's a space missing here.

Response: Yes, we have revised clearly as.

「The pH of the filtrate was measured using a glass-electrode pH meter, and the ammonia nitrogen (NH₃-N) content was determined using the phenol–hypochlorite method based on the previously published method (Broderick & Kang, 1980).」

22. Line 171 - 203: Provide a short description of the determination of organic acids. Also, a description of the assessment of the microbial population is necessary.

Response: Yes, we have revised clearly as.

「The concentrations of organic acids (lactic acid (LA), acetic acid (AA), propionic acid (PA), and butyric acid (BA)) in the oat silage were determined by the high-performance liquid chromatography according to the previously reported method (You et al., 2021). The microbial populations, including LAB, yeasts, and molds, in the FM were counted using the plate count method and expressed as colony-forming units (cfu)/g of FM (You et al., 2021). The de Man, Rogosa, Sharpe agar (Difco Laboratories, Detroit, MI, USA) was used to count the numbers of

LAB after incubating at 30°C for 48 h. The potato dextrose agar (Nissui Ltd., Tokyo, Japan) was used to count the numbers of molds and yeast after incubating at 30°C for 48 h.」

23. Line 215-228: The polymerase chain reaction (PCR) amplification how to performed ?

Response: Yes, we have revised clearly as.

「The polymerase chain reaction (PCR) amplification was performed in a total volume of 25 μ L reaction mixture, including template DNA (25 ng), PCR premix (12.5 μ L), primer (2.5 μ L of each primer), and the volume was adjusted with PCR-grade water. The PCR conditions to amplify the prokaryotic 16S fragments consisted of an initial denaturation for 30 s (98 °C); 35 cycles of denaturation for 10 s (98 °C), annealing for 30 s (54 °C /52 °C), extension for 45 s (72 °C); and then final extension for 10 minutes (72 °C). The PCR products were performed with 1% agarose gel electrophoresis. In the DNA extraction process, ultrapure water was used to exclude the possibility of false-positive PCR results as a negative control. The PCR products were purified using AMPure XT beads (Beckman Coulter Genomics, Danvers, MA, USA) and quantified using Qubit (Invitrogen, USA). The libraries were sequenced either on 300PE MiSeq runs and one library was sequenced with both protocols using the standard Illumina sequencing primers, eliminating the need for a third

index read.」

24. Line 220: There's a space missing here.

Response: Yes, we have revised throughout the manuscript.

25. Line 260 - 267: Show how the difference between treatments was identified. E.g. a ANOVA.

Response: Yes, we have added.

「All data were analyzed using general linear models with the following equation: $Y_{ij} = \mu + \alpha_i + \beta_j + (\alpha \times \beta)_{ij} + \epsilon_{ij}$, where Y_{ij} represents the response variable, μ is the overall mean, α_i represents the effect of the additive, β_j represents the effect of the ensiling period, $(\alpha \times \beta)_{ij}$ represents the interaction effect between the additive and ensiling period, and ϵ_{ij} represents the residual error and the significant differences among treatments were at the 5% probability level (Steel and Torrie, 1980).」

Steel, R.G.D., Torrie, J.H., 1980. Principles and procedures of statistics: A Biometrical Approach, 2th. ed. Mc Graw-Hill Book Co., New York, NY, USA.

26. Line 277: In the results, the description of some of the results analyzed in the manuscript is not very accurate and needs to be narrated

in strict accordance with the results of significance. It is recommended that in the results analysis, all test treatments are represented by abbreviated symbols, which is more concise and clear.

Response: Yes, we have checked revised clearly throughout the manuscript.

27. Line 281 and elsewhere: There should be a ($p > 0.05$) here.

Response: Yes, we have revised throughout the manuscript.

28. Line 346: The "Erwinia" and "Pseudomonas" should be italic.

Response: Yes, we have revised in the manuscript.

29. Line 359-367: Please describe the object of comparison.

Response: Yes, we have revised in the manuscript.

「After 10 days of fermentation, no significant ($p > 0.05$) differences were found in the LP treatment compared to that in the CON and LB treatments. In particular, the genus *Enterococcus* were abundant in the CON and LB treatments. After 60 days of fermentation, no significant ($p > 0.05$) differences were found in the LP treatment compared to these in the CON and LB treatments.」

30. Line 342 and elsewhere: Please standardise the header format throughout the text. "Fig. 1." or "Figure 1".

Response: Yes, we have revised throughout the manuscript.

31. Line 401 and elsewhere: The "*Aspergillus*" should be italic.

Response: Yes, we have revised in the manuscript.

32. Line 405-413 This sentence should be rewrite.

Response: Yes, we have revised in the manuscript.

「Following 7 days of ensiling, the fungal communities within the LB and LP treatments exhibited notable diversity. By the 10th day of fermentation, *Erythrobasidiaceae* became dominant in both the LB and LP treatments, accompanied by an enrichment of various other fungal species. After a 90-day fermentation period, *Apiotrichum* became prominent in both LB and LP treatments. However, it's worth noting that less desirable fungi, such as *Aspergillus*, were more prevalent in the LB treatment.」

33. Line 433: NDF: rho = 0.607777572249648, p < 0.01). Please retain three decimals.

Response: Yes, we have revised in the manuscript.

34. Line 445 and elsewhere: The "Lactobacillus" and "p" should be italic.

Response: Yes, we have revised in the manuscript.

35. Line 466-472: What dynamics of oats.

Response: Yes, we have revised in the manuscript.

「In this study, a combination of multiple physicochemical analyses and high-throughput sequencing was employed to investigate the dynamics of fermentation quality and microbial community of oat silage during anaerobic fermentation with or without inoculation of *L. buchneri* or *L. plantarum*.」

36. Line 511: can → could.

Response: Yes, we have revised in the manuscript.

37. Line 517: Please insert a comma before "and".

Response: Yes, we have revised in the manuscript.

38. Line 641-643: I do not understand the link of this with the oat silage.

Response: There was a mistake. We have revised in the manuscript.

「The fungal community also plays an important role in the development of physical characteristics in silages.」

39. Line 635: "AAAA" ?

Response: Yes, we have revised in the manuscript.

40. Table 2 and 3: There are some significant interaction that are not described in this section.

Response: Yes, we have described and added in the manuscript.

「The WSC, CP, ADF, NDF, and lignin contents were significantly ($p < 0.05$) influenced by the effects of inoculants or ensiling days. The interaction between inoculants and ensiling days significantly ($p < 0.05$) affected the WSC, CP, ADF, NDF, and lignin.」 & 「The LA, AA, PA, and $\text{NH}_3\text{-N}$ contents and pH value were significantly ($p < 0.05$) influenced by the effects of inoculants or ensiling days. The interaction between

inoculants and ensiling days significantly ($p < 0.05$) affected the LA, AA and $\text{NH}_3\text{-N}$ contents.」

41. Figures 1 -2: which can not be read: the letters are too small

Response: Yes, we have revised the letters.

42. Suggest one reference to cite and check: Using PICRUSt2 to explore the functional potential of bacterial community in alfalfa silage harvested at different growth stages

Response: Yes, we have checked and cited in the manuscript.

Reviewer #2

1. Line 271-274 The description of microbial counts of fresh oat was not in line with Table 1.

Response: Yes, we have revised in the manuscript.

「The oat contained low desirable LAB (4.57 log cfu/g of FW) and high mold (4.13 log cfu/g of FW) counts.」

2. Line 136-139 Why the DM content of oats was so high? If you wilted them, the wilt condition need to be illustrated.

Response: Yes, we have revised clearly as:

「Forage oat (*Avena sativa* L.), obtained from the Hulunber Grassland Ecosystem National Observation and Research Station of the Chinese Academy of Agricultural Sciences in Inner Mongolia, China, was harvested at the milk stage of maturity. Then, the oat was wilted for 3 h after harvest under the sunny condition. The samples were collected from three randomly selected sites.」

3. Line 286-292 There were conflicts between 'There were no significant differences in the ADF concentration throughout all treatments.' and 'The ADF concentration was significantly decreased in L. plantarum-treated oat silages compared to both the CON and L. plantarum treatment.'

Response: Yes, we have revised clearly as:

「There were no significant ($p > 0.05$) differences in the ADF concentration in the CON and LP treatments, whereas the ADF concentration was significantly ($p < 0.05$) decreased in the LP treatment compared to that in the CON and LB treatments after 90 days of fermentation」

4. Line 516-518 The reason that *L. buchneri* had a higher pH compared to the CON should be explained.

Response: Yes, we have revised clearly as:

「Therefore, the significant higher pH was observed in the LB group than that in the CON group could be explained by the LA was breakdown into AA and PA by the *L. buchneri*.」

5. The resolution of figures used in the manuscript too low to be clear reviewed.

Response: Yes, we have revised in the manuscript.

6. Line 488-493 was wrong.

Response: Yes, we have revised clearly as:

「Despite meeting the WSC content requirements, the presence of lower LAB counts ($< 5.0 \log \text{cfu/g}$ of FW) may lead to undesirable fermentation and end products (Wang et al., 2022).」

September 27, 2023

Dr. Shuai Du
Zhejiang University
Yuhangtang Road 866
Hangzhou
China

Re: Spectrum02228-23R1 (Physicochemical characteristics and microbial community succession during oat silage prepared without or with *Lactiplantibacillus plantarum* or *Lentilactobacillus buchneri*)

Dear Dr. Shuai Du:

Your manuscript has been accepted, and I am forwarding it to the ASM Journals Department for publication. You will be notified when your proofs are ready to be viewed.

Sincerely,

Jing Han
Editor, Microbiology Spectrum
